# Crop Yield and Essential Oil Composition of Two *Thymus vulgaris* Chemotypes along Three Years of Organic Cultivation in a Hilly Area of Central Italy

**DOI:** 10.3390/molecules26165109

**Published:** 2021-08-23

**Authors:** Basma Najar, Luisa Pistelli, Benedetta Ferri, Luciana Gabriella Angelini, Silvia Tavarini

**Affiliations:** 1Department of Pharmacy, University of Pisa, via Bonanno 6, 56126 Pisa, Italy; basmanajar@hotmail.fr (B.N.); ferribenedetta@live.it (B.F.); 2NUTRAFOOD Interdepartmental Research Center, Nutraceuticals and Food for Health, University of Pisa, via del Borghetto 80, 56124 Pisa, Italy; luciana.angelini@unipi.it (L.G.A.); silvia.tavarini@unipi.it (S.T.); 3Department of Agriculture, Food and Environment, University of Pisa, via del Borghetto 80, 56124 Pisa, Italy

**Keywords:** organic cultivation, chemotypes, thymol, linalool, harvest time, crop age

## Abstract

*Thymus vulgaris* L. is one of the most commonly used medicinal and aromatic plants (MAPs), owing to a range of therapeutic properties of its essential oil. Plant growth, biomass yield, essential oil content and composition are influenced by chemotype, environmental conditions, cultivation techniques and vegetative development. Since in MAPs cultivation special attention is paid on high quality of raw material, the adoption of sustainable agriculture methods is of pivotal importance. Therefore, we evaluated the agronomic and qualitative performances of two *Thymus vulgaris* L. chemotypes, organically cultivated under the Mediterranean climate of hilly lands of central Italy for three consecutive years (2017–2019). Along the trial, total above-ground dry biomass significantly increased from the 1st to 3rd year after planting and large variations in the main biological, biometric and productive traits were observed between the two chemotypes. The ‘thymol’ chemotype EO obviously showed thymol as the major constituent (51.26–49.87%) followed by γ-terpinene and *p*-cymene. The ‘linalool’ chemotype EO showed high percentages of oxygenated monoterpenes (about 90%) with linalool (75%), linalyl acetate (8.15%) and b-caryophyllene (3.2%) as main constituents. This study highlighted that *T. vulgaris* can be successfully organically grown in the hilly lands of Tuscany, with interesting biomass and essential oil yields, even though the plants were in the initial years of crop establishment (start in 2017). The introduction of this species into organic cultivation systems could contribute to obtain high-quality raw material, as well as to enhance crop rotation diversification, which is of pivotal importance in the management of organic farms.

## 1. Introduction

*Thymus* L. genus with about 215 species belongs to the Lamiaceae family, especially distributed in the Mediterranean region [1], and represent one of the most numerous genera for the high number of included species. *Thymus vulgaris* L., commonly known as wild thyme, is a perennial evergreen shrub, low-growing, which performs best in dry, sandy, or rocky soils. It prefers full sun and requires good drainage, with a good tolerance to frost and drought. The species is characterised by different uses: in medicine for the antiseptic activity of the essential oil (antifungal and antimicrobial effect), in cosmetic industry for the typical smell, and as whole plant for culinary purposes as flavouring agent and food additive or preservatives due to its spicy properties [2].

Many compounds with antioxidant properties are reported in thyme and include phenolic acids, flavonoids and essential oils (EOs). Normally the essential oils from *Thymus vulgaris* L. are formed basically by monoterpenes, where the oxygenated derivatives constitute the largest amount. Many papers reported the chemical composition of the thyme essential oils and the similarity of their composition, even though the amount of the main constituents could be often altered, making the commercial application of this oil very difficult [2].

As many other members of the Lamiaceae family from the Mediterranean region (i.e., *Mentha* or *Origanum*), chemical polymorphism can be regarded as an important property of Thymus genera and the intraspecific chemotype variation is particularly striking and probably quite common [3]. This species is known for its six genetically distinct chemotypes that can be distinguished on the basis of the dominant monoterpene produced in glandular trichomes on the surface of the leaves [4]. Each of the six chemotypes are called according to the dominant monoterpene present in the essential oil: carvacrol, geraniol, linalool, α-terpineol, thuyanol-4 (or sabinene hydrate) and thymol [2,3,5,6].The major distinction among these chemotypes is the phenolic nature of carvacrol and thymol and the nonphenolic nature of the other four monoterpenes. In the literature cited in the Thompson work [3], it was suggested that in the EOs of the thymol and carvacrol chemotypes, other monoterpenes co-occurred with relatively high levels (up to 30% of the oil) such as γ-terpinene and *p*-cymene. For geraniol, linalool and α-terpineol chemotypes, the dominant monoterpene occurs at levels up to 90–95% of the oil, whereas the thymol chemotype contains no more than 65% thymol. In addition to the above mentioned chemotypes, another thyme chemotype was detected in Spain, with 1,8-cineole as the main constituent [2], together with two others from natural populations of *T. vulgaris* collected in the Mediterranean region of France, with borneol and *p*-cymene as the main constituents. Satyal et al. [7], on the contrary, following a re-analysis based on enantiomeric analysis, revealed at least 20 different chemotypes. 

Beside chemotype, it is well-known that essential oils and their chemical compositions are strongly affected by environmental conditions and agronomic management practices [8]. The application of sustainable methods of crop cultivation suitable for commercial large-scale production of safe and high-quality herbal material is of pivotal importance. Products originating from conventional farming may not meet all the safety requirements due to the possibility of contamination by nitrate fertilisers and pesticides residues. Thus, organic farming seems a good alternative. Furthermore, in the hilly lands such as those of central Italy, thyme may represent a promising semi-perennial crop for organic agriculture. Its introduction into traditional cereal-based cropping systems, can contribute to their diversification, but also to increase their sustainability. When cultivation is based on an organic and sustainable approach, in fact, it can contribute to the promotion of agro-ecosystem health, including biodiversity, biological cycles and soil biological activity, as well as to the recovery of degraded and marginal lands. Organic agriculture emphasises the use of management practices adapted to local conditions, rather than the use of off-farm inputs. This is accomplished by using, where possible, agronomic, biological and mechanical methods, as opposed to using synthetic materials, to fulfil any specific function within the system (FAO/WHO Codex Alimentarius Commission, 1999). Despite the strong demand for herb organic product, farmers are reluctant to convert their land to organic production. A major barrier is the present lack of information and knowledge about organic farming systems and their effect on yield and quality. In this regard, *Thymus vulgaris* is an example. In fact, although its cultivation is described in the literature [9,10,11,12,13], there are scarce data concerning its organic management [14]. 

The aim of this work was to evaluate the introduction of two *Thymus vulgaris* L. chemotypes (‘thymol’ and ‘linalool’ chemotypes) in organic cultivation in the hilly land of Tuscany (Santa Luce, Pisa Province, Central Italy) and to assess, through three-year field experiment (2017–2019), the effect of crop age on the main biometric and productive characteristics as well as on EO yield and quality. At the same time, the effect of organic management was assessed in terms of soil quality. This work was included in a project financed by the Tuscany Region named ‘*FLORA aromatica Santa Luce e la valle dei profumi: sperimentazione di un modello per la valorizzazione del territori*’ (*Aromatic FLORA Santa Luce and the valley of scents: experimental model for the enhancement of lands and environment-Bando Progetti Integrati di Filiera-PIF-PSR 2014–2020 SOTTOMISURE: 16.2–4.1.3–4.2–6.4.1, CUP 726681)* with the aim to cultivate aromatic plants in organic way to obtain high-quality essential oils, to exploit the local land and marginal areas and to contribute to an ecological and sustainable agriculture.

## 2. Results and Discussion

### 2.1. Growing Season Conditions

Along the experiment, daily air temperature and total rainfall were recorded from the closest weather station (less than 500 m) (Figure 1). For meteorological data, the period from July 2016 to May 2019 was considered and the end of each growing season coincided with the harvest of plant material, occurred from end of April to May in all years of trials (i.e., 11 May 2017 for both chemotypes; 24 April and 11 May 2018 for linalool and thymol chemotype respectively, and 31 May 2019 for both chemotypes).

The climate is typical of the North-Mediterranean area, characterised by a long-term average annual rainfall of 824 mm and a mean annual temperature of 14.3 °C. The thermo-pluviometric trend is characterised by rainfall distributed mainly in autumn (from September to December) and in spring (from March to May), followed by a summer (July-mid-August) drought period with high temperatures. Typically, July is the warmest month (23 °C average monthly temperature) while January the coldest one (6.6 °C average monthly temperature). Such climatic conditions are suitable to thyme growth, which is characterised by a high light and temperature requirements, given its Mediterranean origin. Furthermore, thyme is adapted to alternative dry and rainfall periods [10] and reasonably survives to arid conditions [15].

In the first year of cultivation (July 2016–May 2017) a total rainfall of 742.4 mm, mainly concentrated between September and November and in February, was recorded; November was the wettest month ever, with high intensity rainfall often concentrated in short periods of time. The lowest temperatures were recorded in January (5.6 °C average monthly temperature), the highest in July and August (24.2 °C average monthly temperature), with an average annual temperature of 15.0 °C.

The second year of experimentation (period June 2017–May 2018) was characterised by a total rainfall of 664.2 mm, concentrated between November and March. The minimum temperatures were recorded in February (5.8 °C average monthly temperature), with an average annual temperature of 15.5 °C. In the third year of cultivation (June 2018–May 2019), total rainfall was equal to 533.0 mm, mainly concentrated in autumn (October and November) and spring (April and May). The coldest month was January (5.7 °C) and the average air temperature of the entire period was of 15.6 °C.

### 2.2. Biological, Biometric and Productive Characteristics

Along the three years, total above-ground dry biomass significantly increased from the 1st to the 3rd year after planting and large variations in the main biological, biometric and productive traits were observed between the two chemotypes (Table 1). Taking into account the plant density, the results suggested that the thyme plants, for both chemotypes, were characterised by a good winter survival in the tested environment. In fact, no reduction in plant number per m^2^ has been detected passing from the 1st to 3rd year of cultivation. ‘Thymol’ chemotype was characterised by the tallest plants, with the highest above-ground yield in the 1st and 2nd year, with mean values ranging from 0.20 to 0.88 Mg ha^−1^. In the 3rd year of cultivation, a significant increase in productive performance of ‘linalool’ chemotype has been observed with a total above-ground biomass of 2.94 Mg ha^−1^ on dry weight basis. As general trend, in the first and second year after planting, the ‘linalool’ chemotype was characterised by slower and reduced growth, with fewer number of branches and inflorescences per plant, in comparison with ‘thymol’ chemotype. These high morphological and developmental differences between the two chemotypes observed in the 1st and 2nd years of cultivation strongly decreased in the last year and the ‘linalool’ chemotype showed greater productivity in terms of total above-ground biomass. Overall, the biomass increase observed passing from the first to the third year, for both chemotypes, was expected since the crop generally reaches its maximum production in the third year after planting, thus confirming that, in Central Italy, thyme can be grown as a semi-perennial crop.

The yield reached in the present study were generally lower than or almost equal to that reported in literature, where higher plant densities or conventional cultivation practices were carried out (1.6% and 0.7%, respectively) [11,12,13,14]. In our study, a plant density of 3 plants m^−2^ (with an inter-row spacing of 1.70 m and an intra-row spacing of 0.20 m) has been chosen since the wide inter-row spacing allows an effective mechanical control of weeds and the possibility to introduce cover crops, able to maintain and increase soil organic matter and, thus soil fertility. This agronomic management, based on inclusion in the rotation of cover crops, green manure and use of organic amendments, improved the fertility of the soil in terms of available phosphorus (+106%), organic matter (+8.2%), C/N (+1.9%) and total carbonates (+7.7%), as resulted at the end of the 2nd year of cultivation. It is known as soil organic matter plays an important role in long-term soil conservation and/or restoration by sustaining its fertility, and hence in sustainable agricultural production, due to the improvement of physical, chemical and biological properties of the soil [16]. This is of fundamental importance for the Mediterranean area, where the warm climate and the intensity of cultivation increase the rate of organic matter decomposition thus determining its progressive depletion [17]. At this regard, the study carried out by Lungu et al. [18] showed that organically cultivated sage ensured the conservation of soil fertility and improved the available phosphorus and potassium content. In addition, the introduction of thyme in the hilly and marginal lands, such as those of the present study, can contribute to reduce soil losses due to erosion, nutrient leaching and tillage intensity and frequency, owing to its perennial cycle. Similar functions have also been performed by cover crops which, by covering the soil during winter, contributed to preventing soil erosion by wind and rainwater strength. All together, these agronomic advantages deriving from thyme grown organically have, ultimately, important implications in multifunctional and sustainable agriculture.

### 2.3. EO Yield and Composition

The EO yields of ‘thymol’ chemotype ranged between 1.2% in 2017, 0.56% in 2018 and 0.47% in 2019, while, for the ‘linalool’ chemotype between 2.8%, 1.20% and 1.53 in the three reference years, respectively (Table 1). The strong differences in EO content between the two chemotypes can be due to both the heterogeneity for full flowering that characterised ‘thymol’ chemotype plants and the different leaf to stem ratio, higher for ‘linalool’ chemotype in comparison with ‘thymol’ one. As reported by Andolfi et al. [19], *T. vulgaris* flowers are characterised by the higher EO content, followed by apical leaves and intermediate leaves, while in the stems the EO is present only in traces or completely absent. 

Along the experimentation, the ‘thymol’ chemotype essential oil had a drastic decrease (−58%) in comparison with what was found during the 1st year. This decrease was also noted in the ‘linalool’ chemotypes (−46%), even though a slight increase was seen in 2019 with respect to 2018. The strong decrease observed for both chemotype passing from the 1st to the 2nd and 3rd year of cultivation can be due to the differences in total rainfall and distribution that characterised the studied period. Comparing rainfall among years, 2017 was characterised by higher rainfall (742 mm), followed by 2018 (664 mm) and then 2019 (533 mm). In the rainiest conditions, the EO strongly increased, suggesting that greater or lesser rainfall coincided with the greater or lesser EO content. In this regard, Herraiz-Peñalver et al. [20] found EO yield variations in *Lavandula latifolia* Medik., along two years of cultivation, depending on the differences in annual climatic conditions. Similar findings were obtained by Fernández-Sestelo and Carrillo [21], who found a significant and positive correlation between rainfall and OE yield of spike lavender in different bioregions in Spain.

Taking into account EO yields obtained by steam distillation (SD) at farm level in 2018 and 2019, they ranged between 0.55 and 0.60% for ‘thymol’ chemotype and equal to 1.33% for ‘linalool’ chemotype. 

An overview of the chemical profile in the EO of each *Thymus vulgaris* chemotypes, expressed as the relative percentage of the singular compounds, is summarised in Table 2. The EO from ‘thymol’ chemotype revealed an increase of the number of identified peaks even though it was not statistically significant, because no variance was noted in the percentage of the total identified fraction (Table 3) which exceeded 99%. As the name implies, the ‘thymol’ (36) chemotype was rich in this compound and it represented at least the half of the total composition. The percentage of thymol slightly decreased in the last year (2019) in comparison with the previous two. Thymol was followed by carvacrol (37) (11.7%), γ-terpinene (17) (11.2%) and *p*-cymene (13) (7.2%) in the sample collected in 2017. These constituents were also the most representative ones in the subsequent years despite this order was not respected. In every 2018 and 2019 sample, *p*-cymene overcame the amount of carvacrol which underwent a drastic and statistically significant decrease. In the same ways, γ-terpinene had a radical lowering (from 11.2% in 2017 to 4.5% in 2019) as well as α-terpinene (12) and β-myrcene (9) with a rate of decrease of 66.7% and 58.8%, respectively. On the contrary, borneol (25) and caryophyllene oxide (56) rates had a prominent increase (3 and 11.5 folds, respectively). Other compounds were evidenced only in the first year after collection such as *cis*-linalool oxide (20) only in 2017 and *cis*-sabinene hydrate (18) only in 2018. The EO obtained by steam distillation (SD) showed the same behaviour where, in both years of harvesting, thymol was the main compound even though was present in almost halved percentage compared to the HD technique. *p*- cymene was the second main compound for both years with an amount around 20% followed by γ-terpinene (19.8% and 12.9% in 2018 and 2019, respectively). The last technique (SD) was characterised by the high number of identified picks (38 and 40 picks, respectively). Tricyclene (1%) was found only by SD in 2018 as well as other compounds with lower percentage, which swing between 0.3% (*trans*-cadina-1(2),4-diene (55)) and 0.1% (such δ-3-carene (11), (E)-β-ocimene (16), carvacrol acetate (39), geranial (34)). On the contrary, the identified compounds in 2019 mostly belong to sesquiterpenes such as valencene (50) (0.3%) and β-bourbonene (42) (0.2%).

The two-way PERMANOVA test (Table 3) performed on the EO composition of all samples underlined a significant difference both between the used distillation techniques and the harvesting year. The one-way PERMANOVA done on the composition of the EOs harvested in different years and distilled with both techniques was assessed separately and only those distilled in our laboratory confirmed the significant difference between the years even though the pairwise test was unable to evidence which year was different from the others. SIMPER analysis conferred this difference especially to *p*-cymeme (18.5%), carvacrol (14.0%), γ-terpinene (12.6%) and thymol (11.0%) (Table 4). All these constituents showed a *p*-value of less than 0.05 criterion except for thymol. 

PC2 axis was the one responsible for the differentiation between the two techniques used for distillation; in fact, samples distilled by SD were in positive loading according to this axis, while those obtained by HD were of negative loading (Figure 2a). Even though no significant difference was proven by the one-way PERMANOVA performed on SD samples, this plot positioned them in two different quadrants: SD-2019 was in the positive loading in both axes PC1 and PC2, while SD-2018 was in the opposite quadrant (negative loading on PC1 and positive on PC2). Regarding the HD samples, both 2017 and 2018 years were placed in the down left quadrant (negative loading in both axes) while the sample harvested in 2019 was in the opposite quadrant (positive PC1, negative PC2). HCA cluster (Figure 2b) confirmed what was reported by PCA and divided into two clusters: cluster A composed only of SD-2019 sample and cluster B regrouped all the other and it was subdivided into two subgroups where the first one (B.1) included all the HD samples collected during the three years and B.2 was composed only by SD-2018 sample. 

The domination of the ‘thymol’ chemotype EO was reported in the literature. South Italian samples were studied by Mancini [23] who reported the prevalence, in all localities, of thymol (46.2–67.5%) which was followed by carvacrol (5.7–7.1%). This was in total agreement with the samples harvested and distilled by HD in 2017. The French *Thymus vulgaris* ‘thymol’ chemotype collected in May was very rich in thymol (47.06%) [7]. These authors, beside thymol, evidenced *p*-cymene as the second major constituent (20.07%) followed by linalool (5.00%). This result was in partial agreement with what was found in the current study, where both 2018 and 2019 samples showed the same main compounds and in the same order, whatever technique was used to extract the EOs. Linalool, instead, was present in halved percentage. The 2017 analysis was completely different because the second main constituent was carvacrol instead of *p*-cymene. Schimidt also [24] analysed French thymus samples cultivated in the previous region (Southern France), revealed that this chemotype showed a lesser amount of thymol (38.8%) and a greater rate in *p*-cymene (24.0%). The differences noticed with the different distillation methods were also reported in the literature [25].

Concerning the ‘linalool’ chemotype, the number of identified peaks was a bit less than what was found in ‘thymol’ chemotype (Table 5). Here again no statistical significance was found in the number of identified compounds even though they increased going forward in time. This chemotype revealed an amount of linalool (15) ranging from 70.2% in 2019 to 75.9% in 2017. On the contrary, linalyl acetate (22), the second major constituent, showed an increase in the last year of experiment (2019) which was around 11% in comparison with 2017. This did not deny its decrease during 2018. This fluctuation was noted in almost all compounds present in a percentage greater than 1% in at least one year such as camphene (0.5 vs. 1.6 vs. 1, in the three year of experiment 2017–2018–2019, respectively), camphor (16) (0.8 vs. 2.7 vs. 2.6%), borneol (17) (0.2 vs. 1.2 vs. 0.8), α-terpineol (19) (1.7 vs. 1.4 vs. 2.0%), geranyl acetate (29) (1.1 vs. 0.8 vs. 1.3%), β-caryophyllene (30) (3.7 vs. 3.2 vs. 3.7%) and germacrene D (33) (1.5 vs. 0.8 vs. 0.9%). The two distillation methods (SD and HD) showed a similar content in both linalool and linalyl acetate from samples harvested in 2019, while in 2018, linalool amounts were different (greater in HD than SD) and the linalyl acetate evidenced an opposite behaviour (SD > HD). 2018 was a weird year with regard to the oil composition extracted by SD. In fact, this year was typified by the presence of thymol (25) and carvacrol acetate (28) with the not negligent amount (4.1% and 1.6%, respectively). 

The two-way PERMANOVA test (Table 6) also here highlighted a significant difference in both the distillation method used and among the year of collection. The one-way PERMANOVA performed separately in the EO composition of HD samples and SD samples deep-rooted the statistically significant difference between the year in HD samples while no significant difference was noted between the year in DS samples. The pairwise test distinguished thyme collected in 2017 from the one collected in 2019. More than half of dissimilarity was due to four compounds (Table 7): linalool (22.5%), linalyl acetate (16.0%), camphor (7.8%) and camphene (2) (4.2%). These components had a *p*-value of less than 0.05 criterion (Table 8). 

In ‘linalool’ chemotype PC1 axis (Figure 3a) distinguished SD-2018 samples from all the others and located in a positive loading along this axis. PC2 differentiated HD-2019 from the other HD samples. HD-2019 was of positive loading on PC2 and negative on PC1. In this same position we also found SD-2019. This could be explained by the similarity in their EO compositions especially in their highest percentage in caryophyllene oxide, α-terpineol and geranyl acetate. Below this quadrant (negative PC1 and PC2) the remaining samples were found. HCA cluster (Figure 3b) distributed samples into two clusters: cluster A with only SD-2018 and cluster B with all the others, subdivided into two subgroups: B.1, uniform group, composed by HD-2017 and HD-2018 samples, and B.2 with the samples distilled by two different techniques but collected in the same year (2019). 

Despite the literature is wealthy of papers on the EO composition of *Thymus vulgaris* plants, the ones related to ‘linalool’ chemotype were lacking. Torras [2] investigated Catalonian thyme samples and found a similar amount of linalool as in the current study (74.5%). Inversely, these authors pointed out 1,8-cineol (13.9%) as the second major constituent and deny the presence of linalyl acetate. This result disagrees with the present work, where 1,8-cineol was present with negligible percentages. Schmidt and co-workers in 2012 [24] highlighted myrcene as the second main compounds (5.5%) and linalool was present in slight lesser amount (68.5%). Myrcene is one of the constituents of the EO in the chemotype studied herein but its amount was very low. In a more recent study [7] linalool and linalyl acetate percentages (76.5% and 14.3%, respectively) were almost in agreement with our results. 

The variability in the chemical composition and quality of thyme EO depend on several factors including climatic, seasonal and geographic conditions [7,8,9,10,11,12,13,14,15,16,17,18,19,20,21,22,23,24,25,26], harvest period [27] the storage conditions [28] and distillation techniques [25,29]. The EO can be obtained in different ways: hydrodistillation (HD), the most common conventional method, used in research laboratories, is one of the easiest and routine methods and the most recommended by Pharmacopeia [30]. Steam distillation (SD) is the most widely used to produce EOs on a large scale, because it takes less time and allows better oil recovery [31,32]. Several studies investigated the comparison of the aroma profile composition using these two techniques [33,34,35]. The study of Benmoussa [36] evidenced a decrease of carvacrol in the EO of *Thymus vulgaris* ‘carvacrol’ chemotype when SD was performed (76.2%) instead of HD (89.2%). This latter method however revealed a lesser number of identified peaks. Another study was done by the team of professor Wesolowska [25] who reported that thymol (57.14–71.29%) and carvacrol (10.38–20.40%) were the main constituents identified in SD distilled oil, while thymol (41.34–52.23%), carvacrol (10.12–16.73%), *p*-cymene (9.10–12.12%) and γ-terpinene (6.12–10.20%) dominated in the HD oil. In the current work, all these compounds were present in the HD EO from ‘thymol’ chemotype, but in the SD EO the amount of thymol was around two-folds less than that found in HD and the sum of thymol and *p*-cymene cannot reach the thymol level in the hydrodistilled EO.

## 3. Materials and Methods

### 3.1. Site Characteristics

A three-year field experiment (2017–2019) was carried out on an experimental field established in the inland hilly area of the Pisa Province, southern Tuscany, Italy (Santa Luce, 43°28′42″ N, 10°32′42″ E; 160 m above sea level, slope 15%). This study fits within a larger project carried out in an organic farm regarding the introduction of aromatic crops (lavender, lavandin, oregano and thyme) produced according to the biodynamic production method (subject to Regulation EC n. 834/2007 and Regulation EC n. 848/2018).

At the beginning of the field experiment the soil was analysed at 0–30 cm depth to determine: soil texture (international pipette method), pH (H_2_O, 1:2.5), soil organic matter content (Walkley-Black method), total N content (Kjeldhal method), available P (Olsen method), exchangeable K (BaCl_2_ method), conductivity (conductivity meter), C/N ratio (Table 8). The soil texture is loam with low levels of available P and medium content of both N and organic matter content.

### 3.2. Experimental Design, Crop Management and Plant Material

The agricultural area is characterised by traditional farming systems mainly based on rainfed autumn-winter cereals. In 2015, the experimental site was converted to organic production management, in accordance with EC Regulations, to produce aromatic plant species. The conversion began with a fertility building period, the length of which lasted 12 months. In addition to compost fertilization and annual cover crops (winter and summer mixture containing a great variety of species, mainly grass and legumes), biodynamic preparations of specific fermented plant materials according to Demeter standards, were used to enhance soil fertility and microbial diversity and enable more balanced crop growth and development (500 K and Fladen). Cover crops provided soil cover during the winter months, thus preventing soil erosion by wind and rainwater strength, which reduces organic matter content in the long run. In addition, in April 2017, after thyme planting, a cereal-legume mix (*Triticum* spp. and *Vicia faba* minor L.) was sown as cover crop in the inter-row spacing and, during thyme cultivation, biodynamic preparations have been periodically distributed to the soil. The planting of thyme was carried out in November 2016 by a semi-automatic transplanter. The area was divided in two plots of 2300 m^2^ for each chemotype. The plots were checked for uniformity in terms of soil physical-chemical characteristics, soil moisture, as well as management practices received. Plants were purchased from a local nursery, specializing in the production of aromatic and medicinal plants with organic certification. Well-rooted plantlets (10 cm high), obtained from stem cuttings to avoid plant genetic variability in terms of both morphological and phytochemical traits, were planted adopting a plant density of 3 plants m^2^ with an inter-row spacing of 1.70 m and an intra-row spacing of 0.20 m. Each plot consists of 13 rows, 100-m long with 500 plants in each row and is divided into four randomised replicates.

From the second year onwards, mechanical weed control was performed among rows, while manual weeding was carried out on the row. No pests and diseases have been recorded. Plant survival at the end of winter was monitored and measured.

### 3.3. Biological, Biometric and Productive Characteristics

The plants of each chemotype were collected in the 1st, 2nd and 3rd year after planting (2017, 2018 and 2019, respectively) at the full flowering stage, which occurred from end of April to May in all years of trials (i.e., 11 May 2017 for both chemotypes; 24 April and 11 May 2018 for linalool and thymol chemotype respectively, and 31 May 2019 for both chemotypes) on four fully randomised sampling areas (10 m^2^ each). The main biometric parameters were evaluated on 25 plants for each sampling area. Full flowering was considered when 75% of inflorescences were open.

The following parameters were evaluated: percentage of flowering plants; plant height (cm); fresh and dry yield of above-ground biomass (Mg ha^−1^); and essential oil content (%). Dry weight measurements were carried out after drying samples in a ventilated oven at 40 °C until constant weight.

### 3.4. EO Extraction and Analysis

EOs were obtained by hydrodistillation (HD) in Clevenger apparatus according to the methods previously described by Pistelli [37]. Briefly, 50 g of the dried aerial parts of each chemotype were hydrodistilled for 2 h at 100 °C according to the method reported in the European Pharmacopoeia [30]. The distillation was repeated in triplicate for each of the three years of collection. The essential oil yield was determined as the ratio between the weight of oil obtained and the weight of plant material used. The extracted essential oil was dehydrated by anhydrous sodium sulphate and stored in a small opaque flask at 4 °C in freezer at 4 °C and far from light sources until their analyses.

The steam distillation (SD) was performed by FLORA s.r.l. company (FLORA srl, Lorenzana, Pisa, Italy) using stainless-steel equipment (I.M.P. ITALIA, mod. DS 4000, Ansi304) in 2018 and 2019 years starting from all the aerial parts of the two *Thymus* chemotypes, slightly dried outdoors but under covered environment. Each distillation was done for 2 h. Given the low quantity of plant material the first year of planting (2017) and to the unavailability of the stainless-steel distillator (purchased in the second year of project), it was not possible to do the SD in 2017; moreover, no yields of the SD process were measured by the company.

### 3.5. GC-MS Analysis

Gas chromatography–electron ionisation mass spectrometry (GC–EIMS) analyses were performed with a Varian CP-3800 apparatus equipped with a DB-5 capillary column (30 m × 0.25 mm i.d., film thickness 0.25 μm) and a Varian Saturn 2000 ion-trap mass detector. Analytical conditions were as follows: The oven temperature was programmed rising from 60 °C to 240 °C at 3 °C/min; injector temperature 220 °C; transfer-line temperature 240 °C; carrier gas helium He (at 1 mL/min), injection of 1 μL (5% HPLC grade *n*-hexane solution); split ratio 1:30. The acquisition parameters were as follows: full scan; scan range: 35–400 amu; scan time: 1.0 sec. Identification of the constituents was based on a comparison of (i) their retention times (t_R_)with those of the authentic samples and (ii) their linear retention indices (LRIs), determined rel. to the t_R_ of the series of n-alkanes, and mass spectra with those listed in the commercial libraries (NIST 14 and ADAMS 2007) and laboratory-developed mass spectra library built up from pure substances and components of known oils and MS literature data [22,38,39,40,41,42].

### 3.6. Rate of Variation

It is also called rate of evolution, and allows us to calculate the variation between three values in percentage as the increase of percentage of the class of compounds in the EO between the three years of cultivation. To calculate it the following formula was used:variation(in%) = [(Vf − Vi)/Vi)] × 100
where Vi = 2017 value and Vf = 2019 value

### 3.7. Statistical Analysis

Principal component analysis (PCA) and hierarchical cluster analysis (HCA) were performed as reported in a previous study [43]. Briefly, a matrix of correlation was used for the measure of eigenvalues and eigenvectors in PCA analysis where the plot was performed selecting the two highest principal components (PCs). HCA was performed using Ward’s method with squared Euclidian distances. These two analyses were conducted by the JMP software package 13.0.0 (SAS Institute, Cary, NC, USA). Effect of time and distillation technique was statistically assessed by PERMANOVA test as reported by Najar [44] as well as contribution percentage of each constituent in the dissimilarity which was done through SIMPER test.

Agronomic data were subjected to one-way ANOVA analysis to evaluate the effect of chemotype on the main biometric and productive parameters, using the statistical software CO-STAT Cohort V6.201 (2002). Means were separated on the basis of least significance difference (LSD) test only when the ANOVA F-test per treatment was significant at the 0.05 probability level [45].

## 4. Conclusions

Our findings highlighted that thyme represents a promising semi-perennial crop for organic agriculture in the hilly lands of central Italy. Its introduction into traditional cereal-based cropping systems, can contribute to their diversification, but also to increase their sustainability. Owing to organic production, it is possible to generate raw materials and essential oils with high quality and high added value, fulfilling the growing demands of the medicinal plant industry for products with high active substance content.

## Figures and Tables

**Figure 1 molecules-26-05109-f001:**
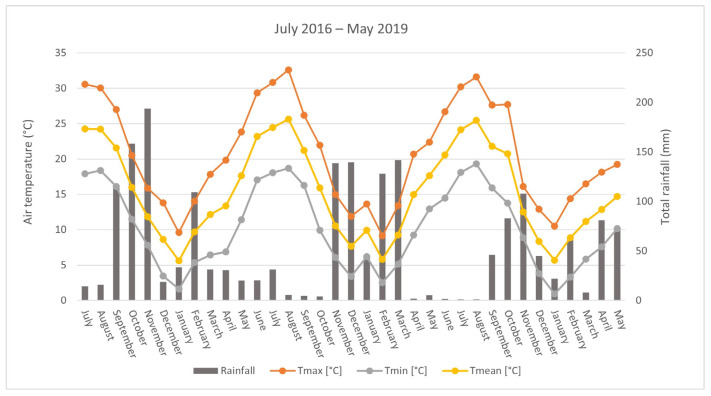
Monthly precipitations and mean minimum, maximum and mean air temperatures (°C) during the study period.

**Figure 2 molecules-26-05109-f002:**
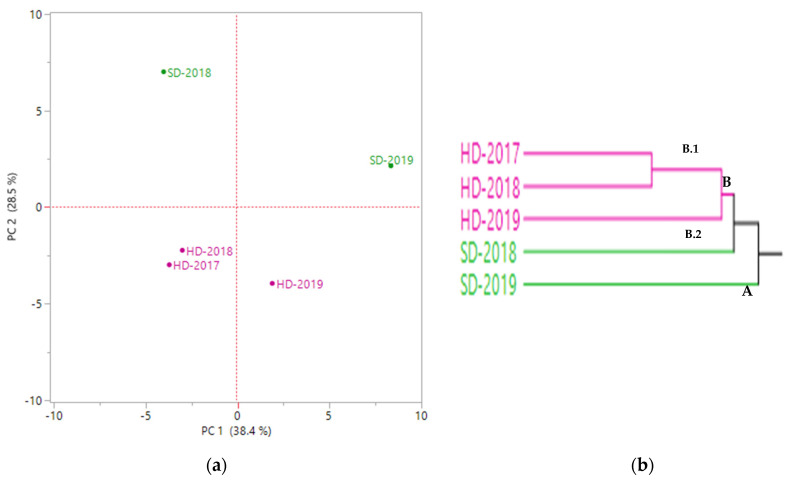
PCA plot (**a**) and HCA cluster (**b**) of the EOs from *Thymus vulgaris* ‘thymol’ chemotype.

**Figure 3 molecules-26-05109-f003:**
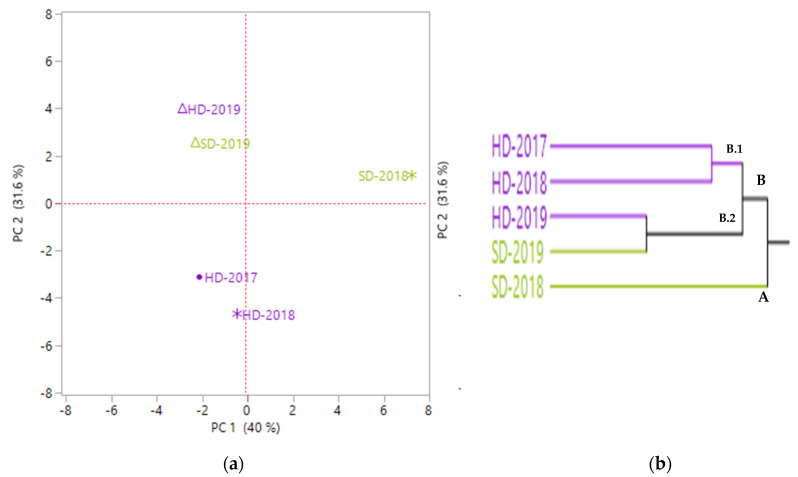
PCA plot (**a**) and HCA cluster (**b**) of the EOs from *Thymus vulgaris* ‘linalool’ chemotype.

**Table 1 molecules-26-05109-t001:** Main biometric and productive characteristics (mean ± SD) of the two *T. vulgaris* chemotypes (‘linalool’ and ‘thymol’) during the three years of growth.

Characters	1st Year ^††^		2nd Year		3rd Year	
	‘Linalool’	‘Thymol’	‘Linalool’	‘Thymol’	‘Linalool’	‘Thymol’
Plant density (n. plants m^−2^)	2.72 ± 0.15	2.65 ± 0.24	2.16 ± 0.45	2.25 ± 0.34	2.94 ± 0.01	2.75 ± 0.17
Plant height (cm)	11.20 ± 0.15 b	17.91 ± 1.66 a	17.15 ± 2.92 b	26.18 ± 3.67 a	22.2 ± 0.69 b	26.33 ± 1.1 a
Total above-ground biomass (Mg FW ha^−1^)	0.23 ± 0.01 b	0.69 ± 0.09 a	1.24 ± 0.39 b	2.93 ± 0.51 a	6.11 ± 0.55 a	3.12 ± 0.30 b
Total above-ground biomass (Mg DW ha^−1^)	0.07 ± 0.01 b	0.20 ± 0.03 a	0.45 ± 0.04 b	0.88 ± 0.09 a	2.94 ± 0.22 a	1.39 ± 0.09 b
EO content (% on dry weight)	2.80 ± 0.23 a	1.20 ± 0.13 b	1.20 ± 0.08 a	0.60 ± 0.12 b	1.53 ± 0.31 a	0.47 ± 0.01 b

One-way ANOVA test, with cultivar as variability factor. For each year, means of each character followed by the same letter are not significantly different at *p* ≤ 0.05 based on LS test. ^††^ Date of harvest 1st year: 11 May 2017; date of harvest 2nd year: 24 April and 11 May 2018 for ‘linalool’ and ‘thymol’ chemotype respectively; date of harvest 3rd year: 31 May 2019.

**Table 2 molecules-26-05109-t002:** Essential oil composition of the ‘Thymol’ chemotype of *Thymus vulgaris* L.

*N*°	Compounds	Class	LRI ^a^	LRI ^b^	Hydrodistillation			Steam Distillation	
					2017	2018	2019	2018	2019
					Relative Abundance (%)				
1	(*E*)-2-Hexenal	nt	854	846	0.4 ± 0.54	0.1 ± 0.09	-	-	-
2	1-Hexanol	nt	867	869 ^§^	-	0.1 ± 0.08	-	-	-
3	Tryciclene	mh	926	921	-	-	-	1.0 ± 0.09	-
4	α-Thujene	mh	931	924	1.0 ± 0.28	0.8 ± 0.04	0.6 ± 0.06	-	0.8 ± 0.06
5	α-Pinene	mh	939	932	0.7 ± 0.23	0.8 ± 0.10	0.7 ± 0.13	1.0 ± 0.01	0.7 ± 0.05
6	Camphene	mh	953	946	0.3 ± 0.26	0.7 ± 0.06	0.9 ± 0.22	0.9 ± 0.09	0.8 ± 0.06
7	1-Octen-3-ol	nt	978	974	0.3 ± 0.02	0.3 ± 0.04	0.3 ± 0.20	0.5 ± 0.03	-
8	β-Pinene	mh	980	974	0.3 ± 0.01	0.3 ± 0.01	-	0.2 ± 0.06	0.4 ± 0.35
9	β-Myrcene	mh	991	988	1.7 ± 0.07	1.1 ± 0.06	0.7 ± 0.10	1.6 ± 0.02	1.0 ± 0.04
10	α-Phellandrene	mh	1005	1002	0.2 ± 0.01	0.2 ± 0.01	-	0.3 ± 0.01	0.2 ± 0.01
11	δ-3-Carene	mh	1011	1008	-	-	-	0.1 ± 0.11	-
12	α-Terpinene	mh	1018	1014	1.8 ± 0.11	1.1 ± 0.09	0.6 ± 0.06	2.4 ± 0.03	1.5 ± 0.04
13	***p*-Cymene**	mh	1028	1020	**7.2 ± 0.70**	**12.6 ± 2.25**	**17.1 ± 2.65**	**20.8 ± 0.21**	**19.3 ± 0.51**
14	Limonene	mh	1031	1024	0.2 ± 0.02	-	0.3 ± 0.04	-	0.4 ± 0.02
15	Eucalyptol	om	1036	1026	1.1 ± 0.06	2.3 ± 0.31	1.1 ± 0.17	1.5 ± 0.96	1.2 ± 0.02
16	(*E*)-β-Ocimene	mh	1050	1044	-	-	-	0.1 ± 0.06	-
17	**γ-Terpinene**	mh	1062	1054	**11.2 ± 0.09**	**6.1 ± 1.47**	**4.5 ± 0.36**	**19.8 ± 2.14**	**12.9 ± 0.20**
18	*cis*-Sabinene hydrate	om	1068	1065	-	1.3 ± 0.22	-	0.8 ± 0.09	0.8 ± 0.01
19	*trans*-Sabinene hydrate	om	1070	1098	-	-	0.7 ± 0.57	-	-
20	*cis*-Linalool oxide (furanoid)	om	1074	1067	1.0 ± 0.19	-	-	-	-
21	Terpinolene	mh	1088	1086	0.1 ± 0.06	-	0.1 ± 0.07	0.2 ± 0.01	0.1 ± 0.01
22	**Linalool**	om	1098	1095	**2.0 ± 0.05**	**2.4 ± 0.39**	**2.6 ± 0.28**	**2.8 ± 0.16**	**2.5 ± 0.06**
23	*cis*-p-Menth-2-en-1-ol	om	1121	1118	-	-	-	0.1 ± 0.03	-
24	Camphor	om	1143	1141	0.4 ± 0.02	0.5 ± 0.10	1.1 ± 1.09	0.9 ± 0.23	1.3 ± 0.03
25	Borneol *	om	1165	1165	0.8 ± 0.12	1.8 ± 0.23	3.4 ± 1.18	2.0 ± 0.02	2.0 ± 0.03
26	4-Terpineol	om	1177	1174	0.5 ± 0.3	0.5 ± 0.09	0.7 ± 0.35	0.7 ± 0.05	0.6 ± 0.02
27	*p*-Cymen-8-ol	om	1183	1179	-	-	0.1 ± 0.10	-	-
28	α-Terpineol	om	1189	1186	0.2 ± 0.02	0.4 ± 0.36	0.3 ± 0.12	0.3 ± 0.9	0.1 ± 0.01
29	*cis*-Dihydro carvone	om	1193	1191	-	0.1 ± 0.01	-	0.1 ± 0.05	-
30	Methyl thymol ether	om	1235	1232	0.7 ± 0.10	0.1 ± 0.01	0.3 ± 0.05	1.2 ± 0.28	1.4 ± 0.03
31	Carvenone	om	1242	1255	-	-	-	0.1 ± 0.04	-
32	Methyl carvacrol ether	om	1244	1241	1.1 ± 0.06	0.3 ± 0.21	1.5 ± 0.94	1.6 ± 0.06	1.9 ± 0.04
33	Geraniol	om	1255	1249	-	-	0.1 ± 0.10	0.2 ± 0.05	0.2 ± 0.01
34	Geranial	om	1270	1264	-	-	-	0.2 ± 0.73	-
35	*Iso*bornyl acetate	om	1285	1283	0.1 ± 0.03	-	-	-	0.5 ± 0.21
36	**Thymol**	om	1290	1289	**51.3 ± 1.77**	**53.7 ± 0.74**	**49.8 ± 4.96**	**26.9 ± 0.90**	**29.8 ± 0.49**
37	**Carvacrol**	om	1298	1298	**11.7 ± 1.11**	**7.7 ± 1.94**	**4.3 ± 1.86**	**4.9 ± 0.04**	**5.6 ± 0.20**
38	thymol acetate	om	1355	1349	-	-	-	-	0.1 ± 0.09
39	Carvacrol acetate	om	1371	1370	-	-	-	0.1 ± 0.12	-
40	α-Copaene	sh	1376	1374	-	-	0.1 ± 0.10	-	-
41	*Iso*bornyl propionate	om	1384	1383	-	-	0.1 ± 0.10	-	0.2 ± 0.01
42	β-Bourbonene	sh	1384	1387	-	-	-	-	0.2 ± 0.01
43	β-Caryophyllene	sh	1419	1417	2.2 ± 0.14	2.8 ± 0.15	2.8 ± 0.36	3.7 ± 0.06	6.4 ± 0.15
44	β-copaene	sh	1429	1430	-	-	-	-	0.1 ± 0.01
45	α-Humulene	sh	1454	1452	-	0.1 ± 0.01	-	0.1 ± 0.04	0.2 ± 0.01
46	γ-Muurolene	sh	1477	1478	-	-	0.3 ± 0.08	-	0.5 ± 0.03
47	Geranyl n-propanoate	nt	1478	1476	0.3 ± 0.13	0.2 ± 0.01	-	0.2 ± 0.01	-
48	Germacrene D	sh	1480	1484	0.3 ± 0.16	0.2 ± 0.09	0.1 ± 0.08	1.1 ± 0.04	1.4 ± 0.02
49	*epi*-Cubebol	os	1493	1493	-	-	0.1 ± 0.10	-	-
50	Valencene	sh	1493	1496	-	-	-	-	0.3 ± 0.01
51	α-Muurolene	sh	1499	1500	-	-	-	-	0.1 ± 0.01
52	β-Bisabolene	sh	1509	1505	-	0.1 ± 0.01	-	0.2 ± 0.11	-
53	γ-Cadinene	sh	1513	1513	-	-	0.2 ± 0.05	-	0.5 ± 0.01
54	δ-Cadinene	sh	1524	1522	0.2 ± 0.02	0.2 ± 0.06	0.4 ± 0.07	0.2 ± 0.06	0.6 ± 0.02
55	*trans*-Cadina-1(2),4-diene	sh	1533	1532 ^§^	-	-	-	0.3 ± 0.07	-
56	Caryophyllene oxide	os	1581	1582	0.2 ± 0.02	0.5 ± 0.09	2.3 ± 0.51	0.5 ± 0.01	2.4 ± 0.03
57	10-*epi*-γ-Eudesmol	os	1619	1622	0.1 ± 0.01	0.1 ± 0.04	-	-	-
58	γ-Eudesmol	os	1632	1630	-	-	0.1 ± 0.10	-	0.1 ± 0.01
59	τ-Cadinol	os	1642	1638	0.1 ± 0.01	0.2 ± 0.16	0.1 ± 0.10	-	0.2 ± 0.01
60	α-Muurolol	os	1646	1644	-	-	-	-	0.1 ± 0.09
61	11,11-Dimethyl-4,8-dimethylenbicyclo[7.2.0]undecan-3-ol	os	1646	1645 ^§^	-	-	0.1 ± 0.07	-	-
62	α-Bisabolol	os	1685	1685	0.1 ± 0.07	-	-	-	-
63	ent-Germacra-4(15),5,10(14)-trien-1B-ol	os	1695	1686 ^§^	-	-	0.1 ± 0.06	-	-
64	Shyobunol	os	1701	1688	-	-	0.3 ± 0.24	-	-
65	Verticiol	os	2106	2106 ^§^	-	-	0.5 ± 0.05	-	-
	Number of identified picks				33	33	38	38	40
	EO Yield				1.2 ± 0.13	0.6 ± 0.12	0.5 ± 0.01	-	-
	Class of Compounds				Hydrodistillation			Steam Distillation	
					2017	2018	2019	2018	2019
	Monoterpene Hydrocarbons (mh)				24.7 ± 0.59	23.7 ± 3.64	25.5 ± 3.54	48.4 ± 1.70	38.1 ± 1,29
	Oxygenated Monoterpenes (om)				70.9 ± 0.71	71.1 ± 2.15	66.1 ± 3.79	44.4 ± 1.10	48.2 ± 0.77
	Sesquiterpene Hydrocarbons (sh)				2.7 ± 0.21	3.4 ± 0.35	3.9 ± 0.21	5.6 ± 0.07	10.3 ± 0.24
	Oxygenated Sesquiterpenes (os)				0.5 ± 0.06	0.8 ± 0.26	3.6 ± 0.16	0.5 ± 0.15	2.8 ± 0.06
	Non-terpene derivatives (nt)				1.0 ± 0.07	0.7 ± 0.23	0.3 ± 0.03	0.7 ± 0.09	-
	Total Identified				99.8 ± 0.20	99.7 ± 0.30	99.7 ± 0.30	99.6 ± 0.40	99.2 ± 0.80

Data are reported as mean values (*n* = 3 ± SD). * Compounds present with a % > 0.1%; Bold type indicates major components; LRI ^a^: linear retention time experimentally determined (HP-5 column); LRI ^b^: linear retention time reported by Adams 2007 [22]; ^§^: linear retention time reported by NIST 2014; *: a significant difference among three years.

**Table 3 molecules-26-05109-t003:** Effects of the harvest year and distillation technique on the EO from *Thymus vulgaris* ‘thymol’ chemotype, according to the two-way PERMANOVA analysis.

Source	F	*p*-Value	Significant Pair-Wise Comparisons at *p* < 0.05
Year	19.91	0.0001	
Distillation Technique	69.259	0.0001	HD versus SD

**Table 4 molecules-26-05109-t004:** Compounds responsible for dissimilarity in the EO from *Thymus vulgaris* ‘thymol’ chemotype according to the SIMPER analysis. *: significant difference between the years (*p*-value < 0.05).

Compounds	Contrib.%	Cumulative %	2017	2018	2019	*p*-Value < 0.05
*p*-Cymene	18.5	18.5	7.2	12.6	17.1	*
Carvacrol	14.0	32.5	11.7	7.7	4.3	*
γ-Terpinene	12.6	45.1	11.2	6.1	4.5	*
Thymol	11.0	56.1	51.3	53.7	49.8	
Borneol	4.6	60.7	0.9	1.8	3.3	
Caryophyllene oxide	4.0	64.7	0.7	0.5	2.3	*
Methyl carvacrol ether	3.3	68.0	1.1	0.3	1.5	
1,8-Cineol	3.0	71.0	1.2	2.3	1.1	
*cis*-Sabinene hydrate	2.4	73.4	0.0	1.3	0.0	*
α-Terpinene	2.3	75.7	1.9	1.1	0.6	*
*cis*-Linalool oxide	2.0	77.7	1.0	0.0	0.0	*
Myrcene	1.9	79.6	1.6	1.1	0.6	*
Champhor	1.8	81.4	0.4	0.5	1.1	
Linalool	1.4	82.8	1.9	2.3	2.6	
β-Caryophyllene	1.3	84.1	2.2	2.8	2.7	
*trans*-Sabinene hydrate	1.2	85.3	0.0	0.0	0.7	
Methyl thymol ether	1.1	86.4	0.7	0.1	0.3	
Camphene	1.1	87.5	0.3	0.7	0.9	
Verticol	1.0	88.5	0.0	0.0	0.5	
α-Thujene	0.9	89.4	1.1	0.8	0.6	
(*E*)-2-Hexanal	0.8	90.2	0.4	0.1	0.0	

**Table 5 molecules-26-05109-t005:** Essential oil composition of the *Thymus vulgaris* ‘linalool’ chemotype.

					Hydrodistillation			Steam Distillation	
	Compounds	Class	LRI ^a^	LRI ^b^	2017	2018	2019	2018	2019
					Relative Abundance (%)				
1	α-Pinene	mh	939	932	0.2 ± 0.01	0.6 ± 0.09	0.4 ± 0.03	0.5 ± 0.01	0.4 ± 0.02
2	Camphene	mh	953	946	0.5 ± 0.01	1.6 ± 0.27	1.0 ± 0.07	1.1 ± 0.07	1.0 ± 0.05
3	1-Octen-3-ol	nt	978	974	0.3 ± 0.01	0.3 ± 0.04	0.5 ± 0.05	0.6 ± 0.03	0.5 ± 0.04
4	β-Pinene	mh	980	974	0.2 ± 0.17	-	-	-	-
5	β-Myrcene *	mh	991	988	0.5 ± 0.02	0.5 ± 0.13	0.3 ± 0.04	0.3 ± 0.01	0.3 ± 0.05
6	*p*-Cymene *	mh	1028	1020	-	0.3 ± 0.08	-	0.2 ± 0.03	-
7	Limonene	mh	1031	1024	0.4 ± 0.02	0.2 ± 0.07	-	0.2 ± 0.01	-
8	Eucalyptol	om	1036	1026	0.2 ± 0.02	0.5 ± 0.22	0.4 ± 0.04	0.8 ± 0.08	0.4 ± 0.02
9	(*Z*)-β-Ocimene	mh	1037	1032	-	-	0.1 ± 0.07	-	0.3 ± 0.03
10	(*E*)-β-Ocimene	mh	1050	1044	0.3 ± 0.03	0.1 ± 0.09	0.3 ± 0.03	-	-
11	γ-Terpinene *	mh	1062	1054	-	-	-	0.6 ± 0.10	-
12	*cis*-Linalool-oxide (furanoid) *	om	1074	1067	0.1 ± 0.01	0.4 ± 0.05	0.5 ± 0.09	0.2 ± 0.01	0.4 ± 0.08
13	Terpinolene	mh	1088	1086	0.2 ± 0.09	0.3 ± 0.14	-	0.3 ± 0.06	-
14	*trans*-Linalool oxide (furanoid)	om	1094	1084	-	-	0.6 ± 0.08	-	0.5 ± 0.06
15	**Linalool**	om	1098	1095	**75.9 ± 0.96**	**75.0 ± 1.20**	**70.2 ± 2.30**	**64.0 ± 1.02**	**71.4 ± 1.27**
16	Camphor	om	1143	1141	0.8 ± 0.02	2.7 ± 0.45	2.6 ± 0.06	2.2 ± 0.08	2.5 ± 0.09
17	Borneol *	om	1165	1165	0.2 ± 0.01	1.2 ± 0.20	0.8 ± 0.02	1.0 ± 0.08	0.7 ± 0.04
18	4-Terpineol	om	1177	1174	-	0.1 ± 0.02	-	0.1 ± 0.04	-
19	α-Terpineol	om	1189	1186	1.7 ± 0.19	1.4 ± 0.03	2.0 ± 0.16	0.3 ± 0.07	2.0 ± 0.18
20	*cis*-Dihydro carvone	om	1193	1191	-	-	-	0.1 ± 0.01	-
21	Nerol	om	1228	1127	0.3 ± 0.04	0.3 ± 0.01	0.4 ± 0.02	-	0.3 ± 0.02
22	**Linalyl acetate**	om	1257	1254	**9.3 ± 0.64**	**6.5 ± 1.04**	**10.3 ± 1.70**	**11.5 ± 0.70**	**10.2 ± 1.58**
23	Isobornyl acetate	om	1285	1283	0.1 ± 0.01	0.4 ± 0.08	-	0.2 ± 0.05	-
24	Bornyl acetate	om	1286	1284	-	-	0.3 ± 0.03	0.4 ± 0.12	0.3 ± 0.02
25	Thymol	om	1290	1289	-	-	-	4.1 ± 0.14	-
26	Carvacrol *	om	1298	1298	-	-	-	1.5 ± 0.14	0.1 ± 0.01
27	Neryl acetate	om	1365	1359	0.5 ± 0.05	-	0.6 ± 0.04	-	0.5 ± 0.03
28	Carvacrol acetate	om	1371	1370	-	-	-	1.6 ± 0.07	-
29	Geranyl acetate	om	1383	1379	1.1 ± 0.09	0.8 ± 0.07	1.3 ± 0.06	0.2 ± 0.01	1.3 ± 0.08
30	**β-Caryophyllene**	sh	1419	1417	**3.7 ± 0.25**	**3.2 ± 0.53**	**3.7 ± 0.31**	**4.7 ± 0.30**	**2.8 ± 0.26**
31	α-Humulene	sh	1454	1452	0.1 ± 0.02	0.3 ± 0.29	0.1 ± 0.08	0.1 ± 0.06	-
32	Geranyl n-propanoate	nt	1478	1476	-	-	0.1 ± 0.06	-	-
33	Germacrene D	sh	1480	1484	1.5 ± 0.11	0.8 ± 0.59	0.9 ± 0.15	1.8 ± 0.23	0.8 ± 0.12
34	δ-Cadinene	sh	1524	1522	-	-	0.1 ± 0.07	0.3 ± 0.07	-
35	α-Cadinene	sh	1538	1537	-	0.1 ± 0.10	-	-	-
36	Geranyl n-butyrate	nt	1562	1562	0.1 ± 0.01	0.3 ± 0.12	0.2 ± 0.18	0.1 ± 0.02	0.2 ± 0.06
37	Germacrene D-4-ol	os	1574	1574	0.6 ± 0.06	0.6 ± 0.45	0.5 ± 0.14	0.2 ± 0.04	0.5 ± 0.12
38	Caryophyllene oxide *	os	1581	1582	0.4 ± 0.04	0.5 ± 0.41	1.4 ± 0.10	0.7 ± 0.05	1.2 ± 0.13
39	*epi*-α-Muurolol	os	1640	1640	-	0.1 ± 0.08	-	-	-
40	τ-Cadinol	os	1642	1638	-	-	0.1 ± 0.08	-	-
41	α-Muurolol	os	1646	1644	0.2 ± 0.03	-	-	-	-
42	11,11-Dimethyl-4,8-dimethylenebicyclo[7.2.0]undecan-3-ol	os	1646	1645 ^§^	-	-	0.1 ± 0.08	-	-
43	α-cadinol	os	1653	1652	-	0.2 ± 0.15	0.2 ± 0.01	-	0.2 ± 0.02
	Number of identified picks				26	28	29	30	24
	EO Yield				2.8 ± 0.23	1.2 ± 0.08	1.5 ± 0.31	-	-
	Class of Compounds				Hydrodistillation			Steam Distillation	
					2017	2018	2019	2018	2019
	Monoterpene Hydrocarbons (mh)				2.3 ± 0.10	3.6 ± 0.50	2.1 ± 0.09	3.2 ± 0.27	2.0 ± 0.12
	Oxygenated Monoterpenes (om)				90.2 ± 0.45	89.3 ± 0.76	90.0 ± 0.96	88.2 ± 1.14	90.6 ± 0.29
	Sesquiterpene Hydrocarbons (sh)				5.3 ± 0.38	4.4 ± 1.04	4.8 ± 0.65	6.9 ± 0.06	3.6 ± 0.37
	Oxygenated Sesquiterpenes (os)				1.2 ± 0.20	1.4 ± 0.63	2.3 ± 0.13	0.9 ± 0.08	1.9 ± 0.03
	Non-terpene derivatives (nt)				0.4 ± 0.01	0.6 ± 0.04	0.8 ± 0.20	0.7 ± 0.03	0.7 ± 0.07
	Total Identified				99.4 ± 0.60	99.3 ± 0.70	99.4 ± 0.60	99.9 ± 0.10	98.8 ± 1.20

Data are reported as mean values (*n* = 3 ± SD). * Compounds present with a% > 0.1%; Bold type indicates major components; LRI ^a^: linear retention time experimentally determined (HP-5 column); LRI ^b^: linear retention time reported by Adams 2007 [22]; ^§^: linear retention time reported by NIST 2014; *: significant differences among three years.

**Table 6 molecules-26-05109-t006:** Effects of the harvest year and distillation technique on the EOs of *Thymus vulgaris* ‘linalool’ chemotype according to the two-way PERMANOVA analysis.

Source	F	*p*-Value	Significant Pair-Wise Comparisons at *p* < 0.05
Year	19.91	0.0001	2017 versus 2019
Distillation Technique	69.259	0.0001	HD versus SD

**Table 7 molecules-26-05109-t007:** Compounds responsible for dissimilarity in the EOs from *Thymus vulgaris* ‘linalool’ chemotype according to the SIMPER analysis. *: significant difference between the years (*p*-value < 0.05).

Compounds	Contrib.%	Cumulative %	2017	2018	2019	*p*-Value < 0.05
Linalool	22.5	22.5	75.9	75.0	70.2	*
Linalyl acetate	16.0	38.5	9.3	6.5	10.3	*
Camphor	8.0	46.5	0.8	2.7	2.6	*
Camphene	4.2	50.7	0.5	1.7	1.0	*
Caryophyllene oxide	4.2	54.9	0.4	0.5	1.4	*
Borneol	3.6	58.5	0.2	1.2	0.8	*
Germacrene D	3.5	62.0	1.5	0.8	0.9	
β-Caryophyllene	3.0	65.0	3.7	3.2	3.7	
α-Terpineol	2.5	67.5	1.7	1.4	2.0	*
trans-Linalool oxide (furanoid)	2.4	69.9	0.0	0.0	0.6	*
Nery acetate	2.2	72.1	0.5	0.0	0.6	*
Carvacrol	2.0	74.1	0.0	0.0	0.5	*
Carvacrol acetate	2.0	76.1	0.0	0.0	0.5	*
Geranyl acetate	2.0	78.1	1.1	0.8	1.3	*
*Iso*bornyl acetate	1.7	79.8	0.1	0.4	0.0	*
Germacrene D-4-ol	1.6	81.4	0.6	0.6	0.5	
α-Pinene	1.6	83.0	0.2	0.6	0.4	*
1,8-Cineole	1.5	84.5	0.2	0.5	0.4	
Limonene	1.4	85.9	0.4	0.2	0.0	*
*cis*-Linalool oxide (furanoid)	1.3	87.2	0.1	0.4	0.5	*
Terpinolene	1.3	88.5	0.2	0.3	0.0	*
Bornyl acetate	1.3	89.8	0.0	0.0	0.3	*
α-Humulene	1.2	91.0	0.1	0.3	0.0	

**Table 8 molecules-26-05109-t008:** Soil characteristic (0–30 cm) at the beginning of the experiment (February 2016).

Parameter	Unit	2016
Sand (2 mm–0.05 mm)	%	39.35
Silt (0.05 mm–0.002 mm)	%	37.75
Clay (<0.002 mm)	%	22.90
pH (1:1 w/v)		8.1
Conductivity	mS cm^−1^	0.159
Total N	‰	1.49
Available P	ppm	3.30
Organic matter	%	2.20
Total carbonates (CaCO_3_)	%	23.78
Active lime (CaCO_3_)	%	12.00
C/N		9.41
Cation-exchange capacity	meq/100 g	11.60

## Data Availability

Data sharing not applicable.

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
