# Peer review of "Crop Yield and Essential Oil Composition of Two *Thymus vulgaris* Chemotypes along Three Years of Organic Cultivation in a Hilly Area of Central Italy"

_molecules, 2021, doi:10.3390/molecules26165109_

Round 1
Reviewer 1 Report
Congratulations to the authors for their research and for the article entitled
„ Crop yield and essential oil composition of two Thymus vulgaris chemotypes along three years of organic cultivation in a hilly area of Tuscany (Central Italy)“
In Keywords add Thymus vulgaris L.
Line 168-168
The yield reached in the present study were generally lower than that reported in literature, where higher plant densities [11-14](11:Khazaie et al., 2008; 14: Kosakowska et al., 2021) or conventional cultivation practices were carried out [11] (11:Khazaie et al.,
should be written
The yield reached in the present study were generally lower than that reported in literature, where higher plant densities [11-14] or conventional cultivation practices were carried out [11]
Line 358 and 382 Adams - reference No [42]
Congratulations to the authors for their research and for the article entitled
„ Crop yield and essential oil composition of two Thymus vulgaris chemotypes along three years of organic cultivation in a hilly area of Tuscany (Central Italy)“
In Keywords add Thymus vulgaris L.
Line 168-168
The yield reached in the present study were generally lower than that reported in literature, where higher plant densities [11-14](11:Khazaie et al., 2008; 14: Kosakowska et al., 2021) or conventional cultivation practices were carried out [11] (11:Khazaie et al.,
should be written
The yield reached in the present study were generally lower than that reported in literature, where higher plant densities [11-14] or conventional cultivation practices were carried out [11]
Line 358 and 382 Adams - reference No [42]
Author Response
In Keywords add Thymus vulgaris L.
Answer: We disagree with the referee because the name of the plant material (Thymus vulgaris L) is already present in the manuscript title and it is well found with a literature search.
Line 168-168: The yield reached in the present study were generally lower than that reported in literature, where higher plant densities [11-14] (11:Khazaie et al., 2008; 14: Kosakowska et al., 2021) or conventional cultivation practices were carried out [11] (11:Khazaie et al.,
should be written:The yield reached in the present study were generally lower than that reported in literature, where higher plant densities [11-14] or conventional cultivation practices were carried out [11]
Answer: the yield found by the cited literature were added. Please see paragraph 2.2. L 171
Line 358 and 382 Adams - reference No [42]
Answer: we corrected and added the reference number below the tables 2 and 5

Reviewer 2 Report
The authors carried out a study on the yields and chemical composition of Thymus vulgaris.
General comments.
References must be cited according to the guidelines for authors.
There are other studies on seasonality and its effects on the yield and chemical composition of essential oil of Thymus vulgaris, what does your work bring new information to the literature that has not yet been explored?
Delete the highlighted markings in the chemical composition table.
Why do three hydrodistillations and only two steam distillations? for example: 2017, 2018 and 2019 HD 2018 and 2019 SD.
What is the calculated RI, based on the series of n-alkane homologues?
How long has steam distillation been carried out?
What is the series of n-alkanes? C8-C40?
Why not using CG-FID for quantification?
Quote the manuscript:
SILVA, Sebastião Gomes et al. Supercritical CO2 extraction to obtain Lippia thyroides Mart. & Schauer (Verbenaceae) essential oil rich in thymol and evaluation of its antimicrobial activity. The Journal of Supercritical Fluids, vol. 168, p. 105064, 2021.
Author Response
General comments.
References must be cited according to the guidelines for authors.
Answer: we corrected the references according the journal guidelines
There are other studies on seasonality and its effects on the yield and chemical composition of essential oil of Thymus vulgaris, what does your work bring new information to the literature that has not yet been explored?
Answer: we agreed with the referee about the presence in the literature of a lot of work on seasonality and its effects on the yield and chemical composition of Thymus vulgaris essential oils. Our work reported the results of both biometric and productive characteristics together with the yield and essential oil composition of Thymus vulgaris organically cultivated along three years of cultivation and this topic is lacking in the literature.
Delete the highlighted markings in the chemical composition table.
Answer: Done
Why do three hydrodistillations and only two steam distillations? for example: 2017, 2018 and 2019 HD 2018 and 2019 SD.
Answer: as reported in line 463-466 paragraph 3.4 given the low quantity of plant material the first year of planting (2017) and to the unavailability of the stainless-steel distillator (purchased by Flora company in the second year of the project), it was not possible to perform the Steam Distillation in 2017.
What is the calculated RI, based on the series of n-alkane homologues?
Answer: The calculated RI is based on the series of n-alkane homologues (C6-C25) on HP-5 column
How long has steam distillation been carried out?
Answer: the steam distillation (SD) was performed for 2 hours according to the FLORA protocol. We reported this time in the paragraph 3.4
What is the series of n-alkanes? C8-C40?
Answer: C6-C25
Why not using CG-FID for quantification?
Answer: The comparisons were made between the same constituents of different samples analysed in the same way, so they are comparable.
Quote the manuscript:
SILVA, Sebastião Gomes et al. Supercritical CO2 extraction to obtain Lippia thyroides Mart. & Schauer (Verbenaceae) essential oil rich in thymol and evaluation of its antimicrobial activity. The Journal of Supercritical Fluids, vol. 168, p. 105064, 2021.
Answer: the authors disagree with the insertion of this paper in the reference of our manuscript since the plant material is different from the two analysed thyme chemotypes reported herein. Furthermore, the supercritical CO2 extraction gave an extract rich in volatiles not comparable with the real essential oils obtained by hydrodistillation or steam distillation of the same plant material.
In fact, according to the "standard ISO 9235, 2013"(recently revised in the standard 6235, 2021) the point 2.11 established that the essential oils are: "Product obtained from a vegetable, natural raw material (2.19), either by steam distillation, by a mechanical process from the peel of citrus fruits or by dry distillation after separation of the aqueous phase, if necessary, by physical processes"

Reviewer 3 Report
This article discusses the possibility of organically cultivating the plant Thymus vulgaris in the region of Tuscany. Cultivation for 3 years gave different results both in the composition and in the analysis of essential oils and other agronomic characteristics such as height, biomass. In addition, certain analyzes were carried out by an external company. While the results are interesting for someone who studies the different clones of this plant, they can only be extended to the region of Tuscany and to specific clones. Under this prism the whole of the work presents partial interests and is not suitable for Molecules. We believe a specialized agricultural or plant physiology journal will be more dedicated to treate this subject.
Author Response
This article discusses the possibility of organically cultivating the plant Thymus vulgaris in the region of Tuscany. Cultivation for 3 years gave different results both in the composition and in the analysis of essential oils and other agronomic characteristics such as height, biomass. In addition, certain analyzes were carried out by an external company. While the results are interesting for someone who studies the different clones of this plant, they can only be extended to the region of Tuscany and to specific clones. Under this prism the whole of the work presents partial interests and is not suitable for Molecules. We believe a specialized agricultural or plant physiology journal will be more dedicated to treate this subject.
Answer: the authors disagree with the referee opinion. Obviously, the results of the three years of organic cultivation are different depending on crop growth and development features of this perennial shrub. It is known that, in thyme, plant stand duration is about 5-6 years and the crop generally reaches its maximum production in the third year after planting. So, the observed increase in biomass yield in the 3 years of cultivation is expected due to this crop behaviour. In addition, the morphological and developmental differences between the two chemotypes, previous observed also by other authors (see for example Kosakowska et al., 2020), determined the different agronomic-productive response during the first 3 years of organic cultivation.
Moreover the chemical analysis were performed both in the university lab to evaluate the EO yield and the composition of the two thyme chemotypes in comparison with the industrial extraction performed on the same plant material but done by the company involved in the cultivation project (FLORA) and responsible for the industrial application of the essential oils obtained in order to exploit the land where the cultivation was done. In our opinion this represents an added value to our work, to select the best thyme chemotype for the industrial cultivation in the selected area of Tuscany. This model can be applied in other areas and with other plant materials.
This journal (Molecules) published other similar papers where organic and conventional cultivation are reported …see these manuscripts as examples:
- “Influence of Agronomic Practice on Total Phenols, Carotenoids, Chlorophylls Content, and Biological Activities in Dry Herbs Water Macerates” https://www.mdpi.com/1420-3049/26/4/1047/htm.
-Characterization of Essential Oil Composition in Different Basil Species and Pot Cultures by a GC-MS Method” DOI 10.3390/molecules22071221
- “Soil and Leaf Nutrients Drivers on the Chemical Composition of the Essential Oil of Siparuna muricata (Ruiz & Pav.) A. DC. from Ecuador” DOI: 10.3390/molecules26102949.
We believe that the agronomic data are supportive to the phytochemical ones and that, therefore, they are a strong point of this paper.
Therefore we believe this manuscript is pertinent to publication in this special issue.

Round 2
Reviewer 2 Report
I believe that most of the recommendations were accepted.